# Construction of room temperature phosphorescent materials with ultralong lifetime by in-situ derivation strategy

Qinglong Jia[1,5], Xilong Yan[1,2,3,4,5], Bowei Wang [1,2,3,4] ✉, Jiayi Li[1], Wensheng Xu[1], Zhuoyao Shen[1], Changchang Bo[1], Yang Li[1,3,4] & Ligong Chen [1,2,3,4] ✉

Although room temperature phosphorescence (RTP) materials have been widely investigated, it is still a great challenge to improve the performance of RTP materials by promoting triplet exciton generation and stabilization. In this study, an in-situ derivation strategy was proposed to construct efficient RTP materials by in-situ deriving guest molecules and forming a rigid matrix during co-pyrolysis of guest molecules and urea. Characterizations and theoretical calculations revealed that the generated derivatives were beneficial for promoting intersystem crossing (ISC) to produce more triplet excitons, while rigid matrix could effectively suppress the non-radiative transition of triplet excitons. Thus, the in-situ derivation strategy was concluded to simultaneously promote the generation and stabilization of triplet excitons. With this method, the ultralong lifetime of RTP materials could reach up to 5.33 s and polychromatic RTP materials were easily achieved. Moreover, the potential applications of the RTP materials in reprocessing or editable anti-counterfeiting were successfully demonstrated.

Long afterglow materials have broad applications in biological imaging[1–3], information encryption[4–6] and optoelectronic devices[7–9], etc., which become the current research hotspot. The traditional superior long afterglow materials are mainly inorganic or organometallic materials[10,11], but these materials normally suffer from inherent defects, such as monochromatic afterglow, complex preparation process and certain cytotoxicity. By contrast, organic room temperature phosphorescence (RTP) has attracted growing attention owing to the low cost, good biocompatibility and high signal-to-background ratio[3,12,13]. As described in Jablonski diagram, it is obvious that the generation and stabilization of triplet excitons are of vital importance for RTP. However, since the intersystem crossing (ISC) from singlet excited states to triplet excited states is a spin-forbidden process, the triplet excitons in organic systems are not easy to be populated. Worse still, the rapid non-radiative decay or quenching of triplet excitons

leads to a significant decrease of phosphorescence emission at room temperature[14,15]. Thus, promoting the generation and stabilization of triplet excitons is a prerequisite for RTP.

Previous research has introduced a variety of innovative methods to regulate the triplet excitons. Some effective routes, such as the heavy-atom effect[16,17], hyperfine coupling[18], and aggregation-facilitated ISC[19,20], were reported to promote ISC processes to produce more triplet excitons. However, these methods required much on the inherent phosphorescence properties or sophisticated molecular designs of the guest molecules[21]. Different from the above methods, the rigid host was confirmed as an effective way to promote guest molecules' RTP emission by inhibiting the non-radiative transition of the triplet excitons, and the reported successful hosts included polymers[6,9,22], organic crystals[23], inorganic matrixes[24–26], macrocyclic molecules[27,28], etc. Although there have been significant advancements

[1]School of Chemical Engineering and Technology, Tianjin University, Tianjin 300350, P. R. China. [2]Zhejiang Institute of Tianjin University, Shaoxing 312300, P. R. China. [3]Tianjin Engineering Research Center of Functional Fine Chemicals, Tianjin 300350, P. R. China. [4]Guangdong Laboratory of Chemistry and Fine Chemical Industry Jieyang Center, Guangdong Province 522000, P. R. China. [5]These authors contributed equally: Qinglong Jia, Xilong Yan. ✉e-mail: bwwang@tju.edu.cn; lgchen@tju.edu.cn

in host-guest RTP materials in recent years, few studies have focused on both the rigid matrix and guest molecules simultaneously[29,30]. Specially, the inherent properties of guest molecules limit the performance of host-guest RTP materials. Thus, it is important to develop a novel strategy to regulate the intrinsic properties of guest molecules for the development of RTP materials during the process of constructing rigid matrix, as it is expected to simultaneously promote the generation and stabilization of triplet-state excitons.

Herein, in-situ derivation strategy was proposed to construct efficient RTP materials via co-pyrolysis of urea and derivable guest molecules, in which the derivatization of guest molecules and the formation of rigid matrix could be achieved simultaneously. Urea was chosen as a pyrolysis precursor because the highly reactive compounds generated during its pyrolysis, such as isocyanic acid and ammonia, could react with guest molecules with specific functional groups. Specially, 1,3,5-tris(4-aminophenyl)benzene (TAPB) was selected as the template molecule and co-pyrolyzed with urea, and the pyrolysis product displayed RTP with lifetime up to 2.43 s. Moreover, material characterization revealed that while the rigid matrix was formed, the derivatives of TAPB were generated attributed to the reaction between TAPB and isocyanic acid. Theoretical calculations indicated that these derivatives played a crucial role in promoting ISC processes to generate more triplet excitons. At the same time, the rigid matrix rich in hydrogen bonds effectively reduced the non-radiative transition of the triplet excitons. As a result, in-situ derivation strategy could simultaneously promote the generation and stabilization of triplet excitons (Fig. 1). According to the above design principles, this strategy had been successfully extended to derivable guest molecules during urea pyrolysis and applied to the preparation of ultralong lifetime or polychromatic RTP materials (Supplementary Fig. 1). The resulting RTP materials showed excellent performance and even an ultralong lifetime of 5.33 s. Furthermore, these materials were applied in reprocessing and editable anti-counterfeiting, attributing to the fact that urea pyrolysis products are good hydrogen bond donors and can form a low eutectic mixture with choline chloride.

## Results

### Photophysical properties for $TAPB_{0.1\%}@U_{30}$

Typically, $TAPB_{0.1\%}@U_{30}$ composite was prepared by one-pot co-pyrolysis of urea and 0.1% mol equiv. of TAPB under 220 °C for 30 min (Fig. 2a). As a comparison, the composite obtained by the separate pyrolysis of urea was named as $U_{30}$. As shown in Fig. 2b and c, $TAPB_{0.1\%}@U_{30}$ exhibited blue luminescence with the peak at 388 nm. The bright green luminescence with emission peak at 519 nm appeared after turning off the UV light and could be captured by naked eyes even lasting for 30 s (Fig. 2b), corresponding to the Commission International del' Eclairage (CIE) coordinates of (0.2957, 0.5001). The blue and

green luminescence were derived from fluorescence and RTP emission, respectively, which could be distinguished by the pronounced disparity in their emission lifetimes[25] (6.38 ns and 2.43 s, Fig. 2d). Notably, the photoluminescence (PL) quantum yield of $TAPB_{0.1\%}@U_{30}$ could reach 15.5%, in which the phosphorescence quantum yield ($\Phi_{ph}$) was 5.37%. As temperature increased, the delayed PL emission intensity gradually decreased and no new emission peaks appeared, indicating that the afterglow emission was entirely from phosphorescence without delayed fluorescence[31] (Fig. 2e). As the temperature increased from 77 to 298 K, the phosphorescence lifetime of $TAPB_{0.1\%}@U_{30}$ decreased from 3.57 to 2.43 s as thermal energy could accelerate triplet exciton non-radiative transition (Supplementary Fig. 2).

The luminescence center of the material was investigated by comparing the photophysical properties of $TAPB_{0.1\%}@U_{30}$ and $U_{30}$. As shown in Supplementary Fig. 3a, significant differences were observed in the phosphorescence emission peaks of $TAPB_{0.1\%}@U_{30}$ and $U_{30}$ at 77 K. Under room temperature conditions, the phosphorescence emission intensity of $U_{30}$ was much weaker than that of $TAPB_{0.1\%}@U_{30}$. Besides, the phosphorescence lifetime of $TAPB_{0.1\%}@U_{30}$ can reach 2.43 s, which was more than 110 times as long as the phosphorescence lifetime of $U_{30}$ (22 ms, Supplementary Fig. 3b). The above results indicated that the phosphorescent emission characteristics of $TAPB_{0.1\%}@U_{30}$ originated from TAPB rather than components produced by the pyrolysis of urea[32]. Compared with TAPB, the red shift of the main emission peaks in the $TAPB_{0.1\%}@U_{30}$ delayed PL and excitation spectra might be attributed to the derivatization of TAPB and the domain limiting effect of hydrogen bond network on guest derivatives in matrix[33–35] (Supplementary Fig. 4). The doping content of TAPB also had a certain impact on RTP emission (Supplementary Fig. 5). When the doping content of TAPB increased from 0.01% to 2.50%, the typical peaks of prompt and delayed PL spectra of $TAPB_x@U_{30}$ remained basically unchanged, but its RTP intensity and lifetime raised at first, then decreased. Notably, when the doping content was 2.50%, the phosphorescence lifetime and $\Phi_{ph}$ were only 1.83 s and 1.96%, respectively (Supplementary Table 1). Therefore, the excessive precursor was not conducive to generate phosphorescent emitters.

### Investigation for RTP mechanism of the $TAPB_{0.1\%}@U_{30}$

In order to better study the mechanism of RTP, it is necessary to explore the composition of RTP materials. $TAPB_{1.0\%}@U_{30}$ was chosen to investigate by high-resolution mass spectrometry (HRMS), because higher TAPB content was beneficial to reflecting the changes of TAPB during co-pyrolysis process. As shown in Supplementary Fig. 6, the peaks at 438.4, 480.5, 523.5, 566.5, 609.6 (m/z) might be attributed to the derivatives derived from the reaction between TAPB and isocyanic acid generated by the pyrolysis of urea. To further confirm this speculation, the mixture of derivatives was extracted from $TAPB_{1.0\%}@U_{30}$ and named as TAPB-N. In the Raman spectrum of TAPB-N, only the characteristic peak of 1363 cm⁻¹ was observed without the peak of triazine ring at 704 cm⁻¹, indicating TAPB-N mainly came from TAPB (Fig. 3a). Compared with TAPB, the chemical shift of protons $H_{a,b}$ has changed obviously in TAPB-N, which might be ascribed to the conjugation effect of the electron-withdrawing carbonyl group. In addition, a series of characteristic peaks were observed in the range of 7.5–8.0 ppm in TAPB-N, attributed to the protons $H_{b,c,d}$ from its derivatives. The characteristic peak of the protons $H_e$ might be attributed to the amide group (Fig. 3b). As for the rigid matrix, the peaks at 129.1, 257.2 and 385.3 (m/z) in HRMS spectrum might be assigned to the pyrolysis products of urea (Supplementary Fig. 6). Powder X-ray diffraction (PXRD) pattern of the composites prepared at different pyrolysis times ($TAPB_{0.1\%}@U_t$, $t$ = 10, 15, 20, 25, 30, 35 min) indicated that the urea pyrolysis transformation pathway involved urea-biuret-ammelide (Fig. 3c, d, Supplementary Discussion 1). The final matrix was confirmed as ammelide according to Fourier transform infrared (FTIR) (Supplementary Fig. 7a), Raman (Supplementary Fig. 7b), and X-ray photoelectron spectra (XPS)

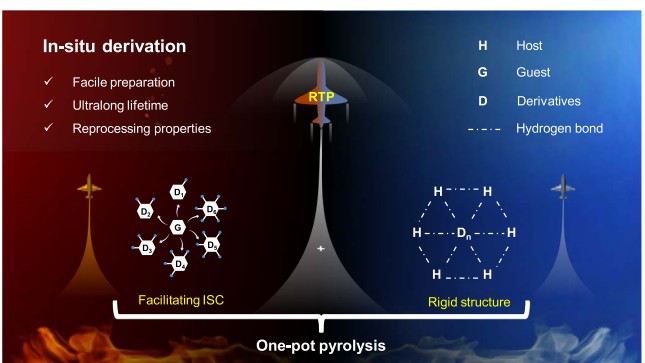

**Fig. 1 | Schematic illustration of the mechanism of in-situ derivation.** Simultaneously promoting the generation and stabilization of triplet excitons by in-situ derivation strategy.

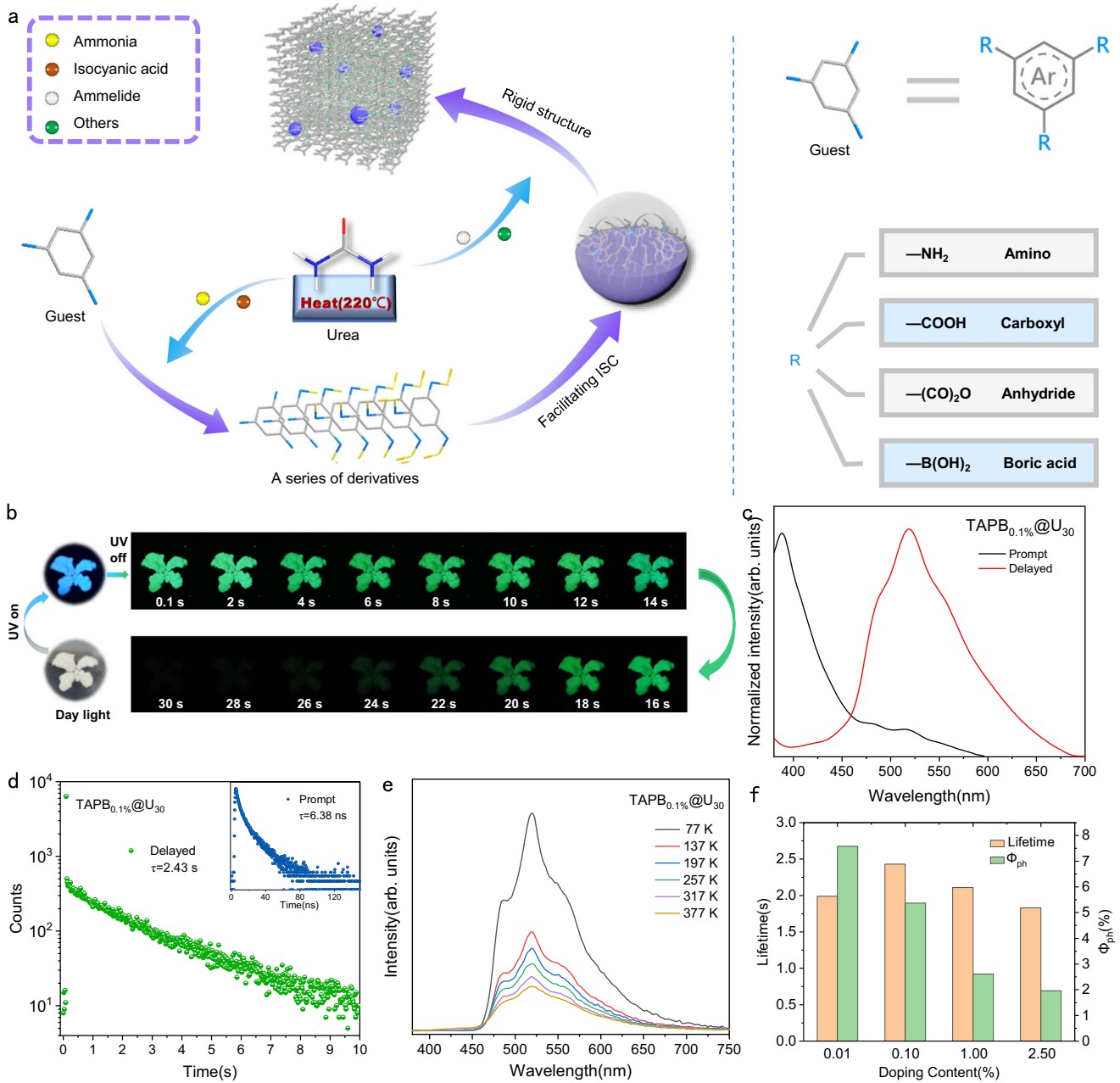

**Fig. 2 | Schematic diagram of the preparation of RTP materials and photophysical properties of TAPB_{0.1%}@U_{30}. a** Preparation of RTP materials by in-situ derivation strategy. **b** Photographs of TAPB_{0.1%}@U_{30} before and after turning off 365 nm UV light at different time intervals. **c** Prompt and delayed photoluminescence (PL) spectra of TAPB_{0.1%}@U_{30}, excited at 365 nm (delay 1 ms). **d** Fluorescence and phosphorescence decay curves of TAPB_{0.1%}@U_{30}, excited at 365 nm and recorded at 388, 519 nm, respectively. **e** Delayed PL spectra of TAPB_{0.1%}@U_{30} at different temperatures (delay 1 ms). **f** Phosphorescence lifetimes and quantum yields of TAPB_{x}@U_{30} ($x$ = 0.01%, 0.10%, 1.00% and 2.50%).

(Supplementary Fig. 8). In Scanning Electron Microscope (SEM) images, the material morphology transformed from fluffy spherical structures to tightly packed lamellar structures (Supplementary Fig. 9). Notably, the RTP lifetime of TAPB_{0.1%}@U_{t} increased significantly with increasing pyrolysis time from 10 to 30 min (Supplementary Fig. 10), indicating that the rigid matrix dominated by ammelide in TAPB_{0.1%}@U_{30} was conducive to RTP. The amount of guest molecule doping was also responsible for the degradation of phosphorescent properties of composites, which could change the crystallization of the material and the composition of the rigid matrix (Supplementary Figs. 11 and 12, Supplementary Discussion 2). As a result, TAPB_{0.1%}@U_{30} exhibited the best phosphorescent performance, the RTP lifetime and $\Phi_{ph}$ were 2.43 s and 5.37%, respectively.

Combined with the control experiments on the photophysical properties for TAPB_{0.1%}@U_{30} and U_{30}, it could be inferred that the production of TAPB-N contributed significantly to high-performance RTP. In order to understand the internal mechanisms underlying the significant RTP performance of the composite, hole-electron analysis was performed to investigate the excitation properties of TAPB and TAPB-N[36–38]. As shown in Supplementary Fig. 13, both S_1 and T_1 states of TAPB exhibited a small degree of hole-electron separation. Similar hole and electron distribution spreading over the whole molecule led to a small spin-orbit coupling (SOC) constant, resulting in undesirable RTP performance. Nevertheless, TAPB-N possessed different properties in the S_1 state. In the case of TAPB-1 (one of the components in TAPB-N, Supplementary Fig. 6), there were significant differences in hole and

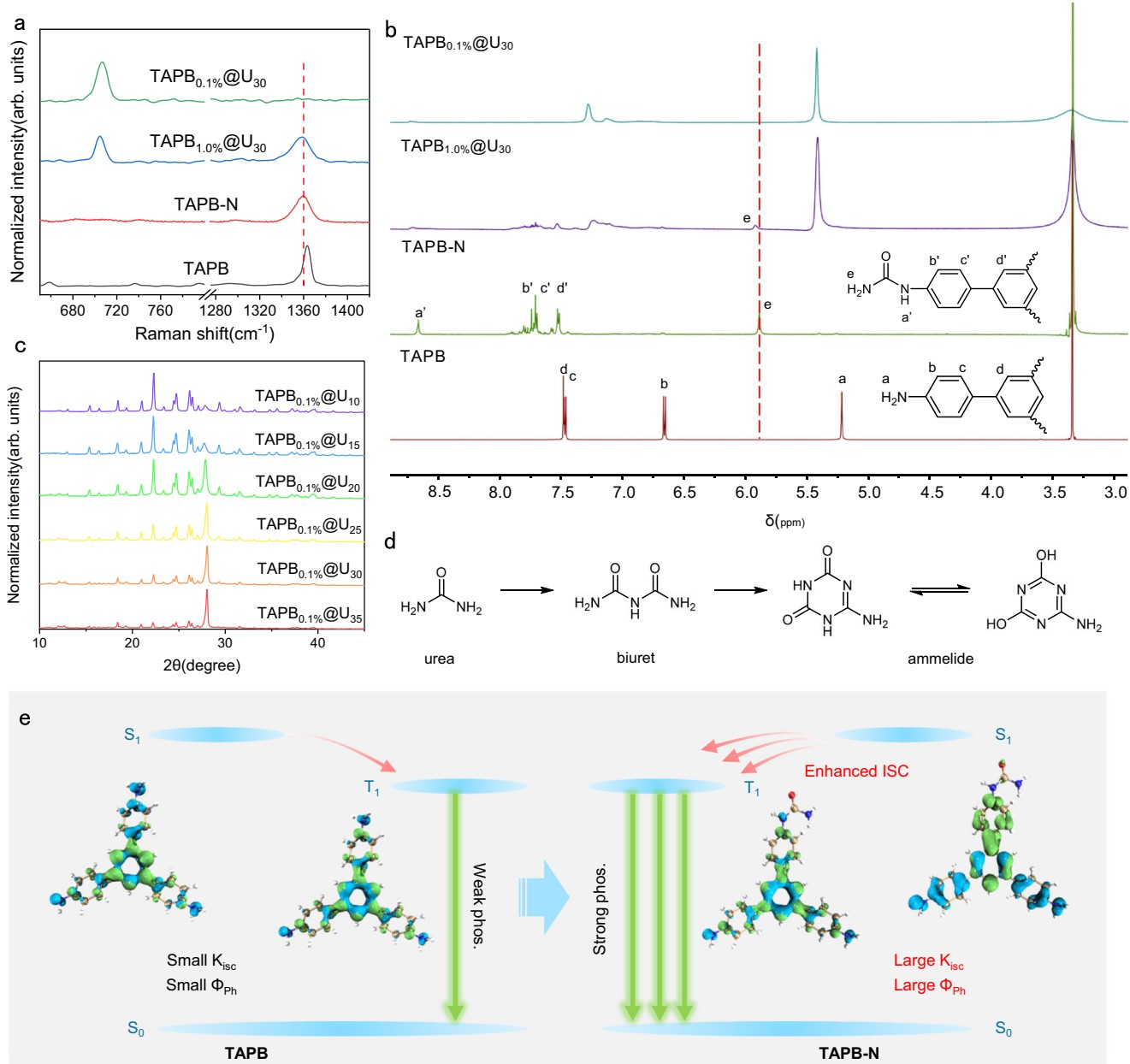

**Fig. 3 | Mechanism investigation of ultralong phosphorescence. a** Raman spectra of 1,3,5-tris(4-aminophenyl)benzene (TAPB), TAPB-N, $TAPB_{1.0\%}@U_{30}$ and $TAPB_{0.1\%}@U_{30}$. **b** $^1$H Nuclear Magnetic Resonance (NMR) of TAPB, TAPB-N, $TAPB_{1.0\%}@U_{30}$ and $TAPB_{0.1\%}@U_{30}$ in DMSO-$d_6$. **c** Powder X-ray diffraction (PXRD) patterns of $TAPB_{0.1\%}@U_t$ ($t$ = 10, 15, 20, 25, 30, 35). **d** Generation pathway of rigid matrix during pyrolysis. **e** Proposed mechanism for enhancing RTP performance (blue and green isosurfaces correspond to hole and electron distributions, respectively).

electron distribution, which might be attributed to the crucial role of the carbonyl group in regulating its excited state electronic structure. Compared with TAPB, the overlap between the electrons and holes in TAPB-1 decreased with the introduction of carbonyl group. Meanwhile, the degree of separation between electrons and holes further increased (Supplementary Table 2). The electrons were more inclined to transfer to the region near the carbonyl group, thereby exhibiting typical intramolecular charge transfer (CT) properties. Compared with locally excited systems, intramolecular CT systems possessed relatively small singlet-triplet splitting energy ($\Delta E_{ST}$), which was crucial to promote ISC and provide more effective ISC channels[3,15,39]. The calculated SOC constants of $S_1$-$T_3$, $S_1$-$T_5$ and $S_1$-$T_6$ were 0.30, 0.58 and 0.30 cm$^{-1}$, respectively (Supplementary Table 3). Large SOC constants could facilitate the ISC process from $S_1$ to $T_n$ and generate more triplet-state excitons, which helped to improve the $\Phi_{ph}$ of the composite.

Similarly, other derivatives of TAPB were examined following the same method, and the intramolecular CT characteristics of $S_1$ state were also confirmed by hole-electron analysis (Supplementary Fig. 13, Supplementary Table 2). Most importantly, the $T_1$ excited state of TAPB-N exhibited LE properties and similar excited state energy compared with $T_1$ state of TAPB, indicating that TAPB-N might have similar phosphorescence emission wavelengths with TAPB. In summary, the generation of derivatives significantly enhanced the ISC process and phosphorescence efficiency without changing the phosphorescent emission peaks of guest molecules (Fig. 3e). To further validate the calculations, PVA@TAPB-N and PVA@TAPB were prepared by doping TAPB-N and TAPB into polyvinyl alcohol (PVA), respectively. Notably, PVA@TAPB-N and PVA@TAPB exhibited similar emission wavelengths and lifetimes (Supplementary Fig. 14). However, PVA@TAPB-N presented a larger $\Phi_{ph}$ than PVA@TAPB, and the intersystem crossing rate

constant ($K_{isc}$) of PVA@TAPB-N was 2.6 times that of PVA@TAPB, which was consistent with the above calculated results (Supplementary Table 4).

Moreover, the rigid matrix dominated by ammelide was also supposed to play an important role in high-performance RTP of the composite. According to thermogravimetric analysis (TGA), $TAPB_{0.1\%}@U_{30}$ showed thermal stability below 150 °C (Supplementary Fig. 15a). Only 5% mass loss was observed with the temperature increase from 150 to 220 °C, which might be caused by the urea or biuret that was pyrolyzed incompletely in the composite. The endothermic peak at 80 °C could be observed in differential scanning calorimetry (DSC) curve, attributed to the hydrogen bond breakage of $TAPB_{0.1\%}@U_{30}$[40] (Supplementary Fig. 15b). The above analysis confirmed the existence of abundant hydrogen bonds in the composite. Taking $TAPB_{1.0\%}@U_{30}$ as an example to verify the interaction between the matrix and guest molecules. It was worth noting that the proton $H_e$ chemical shift of TAPB-N shifted 0.05 ppm to high field due to hydrogen bonds in $TAPB_{1.0\%}@U_{30}$ (Fig. 3b). In the Raman spectrum of $TAPB_{1.0\%}@U_{30}$, the characteristic peak of TAPB-N at 1363 $cm^{-1}$ was widened and shifted 3 $cm^{-1}$ to the low wavenumber, which further confirmed that hydrogen bonds in the matrix can effectively suppress the vibration of guest molecules (Fig. 3a). As a result, the derivatives were firmly fixed in the rigid matrix by hydrogen bonds, effectively inhibiting energy loss caused by non-radiative transition and quenching of triplet excitons. Benefiting from the protection of the rigid matrix, the composite showed good stability in most organic solvents (Supplementary Fig. 16). Notably, the phosphorescence of the composite dispersed in dimethyl sulfoxide (DMSO) was significantly diminished, which might be due to the destruction of hydrogen bond network caused by the dissolution of the materials in DMSO. In addition, the composite was stable in aqueous medium with a range of pH from 2 to 11 (Supplementary Fig. 17), but its phosphorescence exhibited a certain quenching in high concentrations of acid or basic solutions. This might be ascribed to the destruction of the rigid matrix by high concentration of acids or bases, exposing the luminescent center directly to quenching agents such as water or oxygen. This feature was expected to be further used for pH detection.

Based on these experimental results and theoretical calculations, both the generated derivatives and rigid matrix made important contributions to high-performance RTP. Therefore, a possible mechanism for achieving and enhancing RTP emission through in-situ derivation strategy could be proposed. On the one hand, the derivatives of guest molecule generated by co-pyrolysis promoted the separation of excited state electrons and holes, and even exhibited certain intramolecular CT excitation properties. The transformation from guest molecules to derivatives was conducive to promote the ISC process and produce more triplet excitons. On the other hand, the rigid matrix formed by urea pyrolysis contained abundant hydrogen bonds, which could effectively inhibit the quenching and non-radiative decay of triplet excitons. In summary, in-situ derivation strategy successfully promoted the generation and stabilization of triplet excitons.

### Extended experiments on the universality of the design principle

Based on the above discussion, the derivable guest molecules during urea pyrolysis were expected to construct high-performance RTP materials by in-situ derivation strategy. To expand the universality of in-situ derivation strategy and construct more ultralong lifetime and polychromatic RTP materials, benzene-1,4-diboronic acid (P2BA), 1,3,5-tris(4-carboxyphenyl)benzene (TCPB) and 2,3-naphthalenedicarboxylic anhydride (23NDCA) were chosen as template guests for further investigation (Supplementary Fig. 1). Considering the difference of their reactivity with the products of urea pyrolysis, the co-pyrolysis time of the above guests with urea was extended to 60 min.

The bright blue afterglow of about 50 s (lifetime = 5.33 s, $\Phi_{ph}$ = 10.82%) was observed from $P2BA_{0.1\%}@U_{60}$ by naked eyes after turning off the 254 nm UV light irradiation, which was consistent with the typical emission peak at 416 nm in the delayed PL emission spectrum (Fig. 4a, b, Supplementary Fig. 18a). Considering the limitation of microsecond lamps, the lifetime of $P2BA_{0.1\%}@U_{60}$ was further calculated by kinetic attenuation curve measurement, and its 5.02 s ultralong life was far beyond that of most RTP materials (Fig. 4c). Notably, similar delayed emission spectra of $P2BA_{0.1\%}@U_{60}$ at room temperature and P2BA solution ($1×10^{-5}$ M) at 77 K confirmed that the delayed emission of $P2BA_{0.1\%}@U_{60}$ originated from P2BA rather than other emission centers[41] (Supplementary Fig. 19). As the temperature increased from 77 to 137 K, the emission intensity of delayed PL spectra decreased significantly at 417 nm, confirming that it was phosphorescent emission (Fig. 4e). Moreover, the phosphorescence intensity only decreased slightly at 417 nm when the temperature was further increased from 137 to 377 K, which can be explained by the strong confinement effect of the rigid matrix on P2BA. Furthermore, both $TCPB_{0.1\%}@U_{60}$ and $23NDCA_{0.1\%}@U_{60}$ showed strong blue fluorescence under 365 nm UV light with emission peaks at 424 and 426 nm, respectively, which can be confirmed by its shorter luminescence lifetime (Fig. 4a, Supplementary Fig. 20). Green afterglow of about 25 s could be observed after turning off the UV light (Fig. 4b) and confirmed by the typical peaks at 540 and 525 nm in delayed PL spectra, respectively. Their RTP lifetimes calculated from the lifetime curve fitting were 1.70 and 1.52 s and the corresponding $\Phi_{ph}$ were 10.26% and 5.88%, respectively (Fig. 4d, Supplementary Fig. 18, Supplementary Table 5).

Notably, the matrix composition of $P2BA_{0.1\%}@U_{60}$, $TCPB_{0.1\%}@U_{60}$ and $23NDCA_{0.1\%}@U_{60}$ was consistent with that of $TAPB_{0.1\%}@U_{30}$ (Supplementary Fig. 21), which could effectively suppress the quenching and non-radiative transitions of triplet excitons. Taking TCPB as a template molecule, the generation of derivatives during co-pyrolysis was confirmed by HRMS, which showed similar electronic excitation properties to those of TAPB derivatives (Supplementary Fig. 22). Specially, hole-electron analysis confirmed that these derivatives could enhance the separation of excited state electrons and holes, and promote the generation of triplet excitons (Supplementary Fig. 23, Supplementary Table 6, Supplementary Discussion 3). Thermally activated delayed fluorescence (TADF) was also observed in $TCPB_{0.1\%}@U_{60}$ and $23NDCA_{0.1\%}@U_{60}$, which might be closely related to the separation of excited state electrons and holes of derivatives[31,42] (Fig. 4f, g). In summary, in-situ derivation strategy could regulate the intramolecular electron distribution, which was conducive to the construction of high-performance RTP materials. Selecting different derivable guest molecules could endow them with special electronic excitation properties, which would provide some insights for the development of RTP materials and even other long afterglow materials.

In-situ derivation strategy was not only used to construct green or blue RTP materials, but also had good application prospects for the preparation of polychromatic RTP materials. 3-Aminophenylboronic acid (3ABBA), 1-naphthylamine (aNA), 2-naphthalenecarboxylic acid (2NPAC), 1,8-naphthalene-dicarboxylic anhydride (18NDCA), and 1-aminopyrene (1AMP) were chosen to construct colorful RTP composites under the same conditions. As shown in Supplementary Fig. 24, all composites exhibited intense blue fluorescence under 365 nm UV excitation. After the UV light was turned off, the composites showed a sky-blue to red afterglow in turn, with the emission wavelength ranging from 469 to 671 nm (Supplementary Fig. 25). Specifically, the photophysical properties of composites were summarized in Supplementary Table 7. The CIE coordinates of the multicolor materials were described in Supplementary Fig. 26, which further confirmed that the in-situ derivation strategy can easily achieve RTP materials with a wide range of adjustable colors.

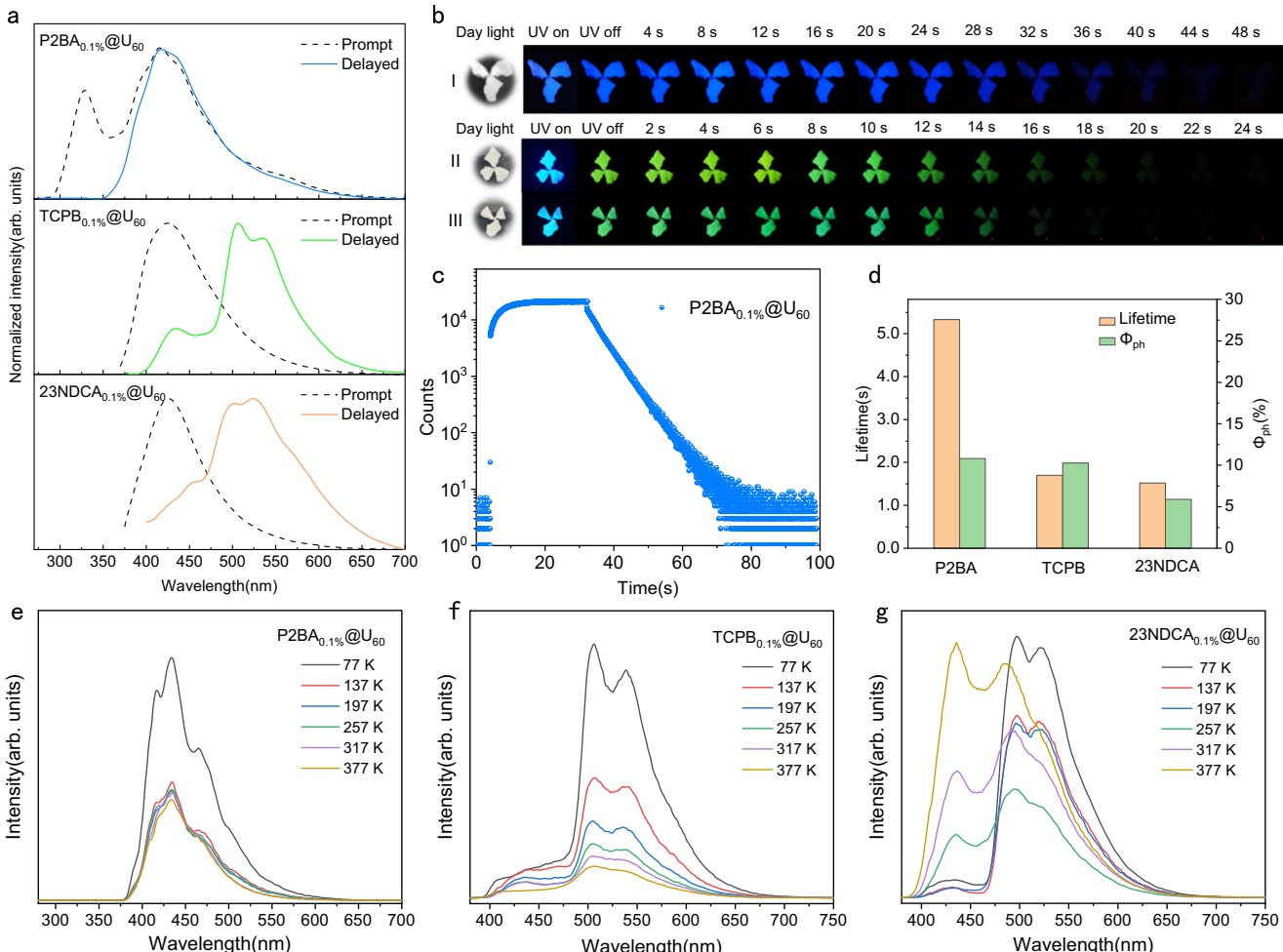

**Fig. 4 | Photophysical properties of P2BA$_{0.1\%}$@U$_{60}$, TCPB$_{0.1\%}$@U$_{60}$ and 23NDCA$_{0.1\%}$@U$_{60}$. a** Prompt and delayed PL spectra of P2BA$_{0.1\%}$@U$_{60}$, TCPB$_{0.1\%}$@U$_{60}$ and 23NDCA$_{0.1\%}$@U$_{60}$ (delay 1 ms). **b** Photographs of P2BA$_{0.1\%}$@U$_{60}$ (I, excited at 254 nm), TCPB$_{0.1\%}$@U$_{60}$ (II, excited at 365 nm) and 23NDCA$_{0.1\%}$@U$_{60}$ (III, excited at 365 nm) before and after turning off UV light at different time intervals. **c** Kinetic decay curve of P2BA$_{0.1\%}$@U$_{60}$. **d** Phosphorescence lifetimes and quantum yields of P2BA$_{0.1\%}$@U$_{60}$, TCPB$_{0.1\%}$@U$_{60}$ and 23NDCA$_{0.1\%}$@U$_{60}$. Delayed PL spectra of P2BA$_{0.1\%}$@U$_{60}$ (**e**) TCPB$_{0.1\%}$@U$_{60}$ (**f**) and 23NDCA$_{0.1\%}$@U$_{60}$ (**g**) at different temperatures (delay 1 ms).

## Applications of RTP materials prepared by in-situ derivation strategy

When conventional crystalline RTP materials were applied in anti-counterfeiting or other fields, they usually showed a poor application prospect due to the high melting point and poor processing properties. RTP materials prepared by in-situ derivation strategy not only solved the processing problem effectively, but also retained the excellent phosphorescent properties of crystalline RTP materials. Based on the low melting point eutectic mixture of choline chloride and urea and its derivatives, the processing properties of RTP materials prepared by in-situ derivation method can be effectively improved by adding choline chloride[43]. The softening temperature of the composites decreased from 220 to 130 °C when 1% mass equivalent of choline chloride was added. As expected, the lower softening temperature made the RTP composites easier to be processed. Notably, the phosphorescent performance of the composite showed good stability with the addition of choline chloride, which is conducive to further processing and application (Supplementary Fig. 27).

Inspired by this, a series of RTP materials for editable anti-counterfeiting with different shapes and colors were prepared by rubber mold (Fig. 5a). The choline chloride and RTP composites were mixed thoroughly and heated until the mixture became a viscous fluid, and the RTP materials with different shapes prepared by pouring the mixture into rubber mold. By assembling different shapes of RTP materials into the desired pattern, polychromatic flowers as well as green and sky-blue leaves could be observed after turning off the UV light. Owing to diverse phosphorescence lifetime of the RTP materials, different flowers were observed at different times (Fig. 5b). It was worth noting that all RTP materials were movable, and they could be arranged into special shapes as desired. In addition, the RTP materials with different shapes still maintained good RTP properties after repeatedly processing more than five times (Fig. 5c). Benefiting from their good processing properties, composites could also be re-processed into other shapes depending on actual requirements. This was expected to provide some insights for the application of conventional crystalline RTP materials.

## Discussion

In conclusion, in-situ derivation strategy has been established to construct efficient RTP materials. This strategy exhibited excellent application prospect for derivable guest molecules during urea pyrolysis. TAPB was utilized as a template molecule to investigate the mechanism of in-situ derivation strategy. Material characterization and theoretical calculations showed that the derivatives generated during the co-pyrolysis process could facilitate the formation of intramolecular charge transfer states and were expected to promote the ISC process. At the same time, rigid matrix composed of ammelide could effectively reduce the quenching and non-radiative decay of the triplet excitons.

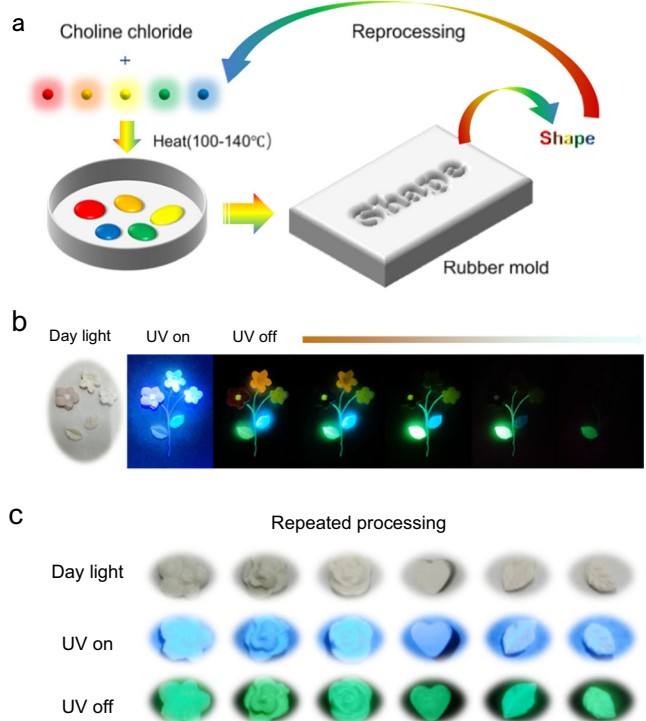

**Fig. 5 | Applications of RTP materials prepared by in-situ derivation strategy.**
**a** Schematic illustration of the application process by adding choline chloride.
**b** Photographs of RTP materials coordinating with choline chloride before and after turning off 365 nm UV light. **c** Photographs of RTP materials after repeated processing.

In a word, the generation and stabilization of triplet excitons for the activation of RTP materials can be achieved simultaneously by in-situ derivation strategy. This strategy has successfully achieved the ultra-long lifetime up to 5.33 s and polychromatic RTP materials. Furthermore, RTP materials prepared in this strategy could not only solve the poor processing properties effectively, but also retained the phosphorescent properties of crystalline RTP materials. This discovery will undoubtedly promote the in-depth understanding of RTP materials and broaden their practical applications.

## Methods

### Reagents and materials
All chemical reagents used in the study were purchased from commercial source without further purification. Urea (purity: 99%), polyvinyl alcohol (PVA) (MW≈20,000, alcoholysis degree: 88%) was purchased from Bidepharm. 1,3,5-Tris(4-aminophenyl)benzene (TAPB) (purity: 98%), benzene-1,4-diboronic acid (P2BA) (purity: 98%), 1,3,5-tris(4-carboxyphenyl)benzene (TCPB) (purity: 98%), 2,3-naphthalenedicarboxylic anhydride (23NDCA) (purity: 95%) and other reagents were purchased from Heowns and used without further purification.

### Measurements
Absolute quantum yields, prompt and delayed PL spectra and phosphorescence decay curves were measured on an Edinburg FLS1000 fluorescence spectrophotometer (Edinburgh Instruments, UK). X-ray diffraction (XRD) analyses were carried out on Bruker AXS D8 X-ray diffractometer (Germany) using a Cu Kα X-ray source (40 kV, 100 mA). Field emission scanning electron microscopy (SEM) was operated on a Hitachi S-4800 microscope. Fourier transform infrared spectra (FTIR) were collected on a Nicolet 380 FTIR spectrometer. Raman spectroscopic studies were carried out on a LabRAM HR Evolution

spectrometer with a 785 nm laser as the excitation source. X-ray photoelectron spectroscopy (XPS) data were recorded by using an X-ray photoelectron spectrometer (K-Alpha +) with an Al Kα X-ray source. The binding energy was calibrated by the C$1s$ peak at 284.8 eV as the reference. High-resolution mass spectra (HRMS) were recorded on a Bruker Daltonics microTOF-QII instrument. Thermogravimetric analysis (TGA) and differential scanning calorimetry (DSC) analyses were performed on a Simultaneous Thermal Analyzer (STA) 8000 with a heating rate of 5 °C·min$^{-1}$. The $^1$H NMR of the sample were analyzed with JEOL JNM ECZ600R at room temperature.

### Theory calculation
TD-DFT calculations were performed on ORCA 4.2.1 program with B3LYP functional and def2-TZVP(-f) basis set to study the photophysical properties of guest molecules[44]. Spin-orbit coupling (SOC) matrix elements between the singlet excited states and triplet excited states were calculated with spin-orbit mean-field (SOMF) methods. The optimized electronic structures were analyzed by Multiwfn software[37,38]. All isosurface maps to show the electron distribution and electronic transitions were rendered by Visual Molecular Dynamics (VMD) software based on the exported files from Multiwfn[45].

### Preparation of RTP composites
Urea (3.0 g, 50 mmol) and TAPB (17.6 mg, 0.05 mmol) were evenly mixed by grinding, and heated at 220 °C for 30 min to afford TAPB$_{0.1\%}$@U$_{30}$ composite. The composite prepared by pyrolysis of urea at 220 °C for 30 min was named as U$_{30}$.

A variety of composites were prepared by co-pyrolysis of urea and derivable guest molecules for 60 min, which were named as P2BA$_{0.1\%}$@U$_{60}$, TCPB$_{0.1\%}$@U$_{60}$, 23NDCA$_{0.1\%}$@U$_{60}$, 1AMP$_{0.1\%}$@U$_{60}$, 18NDCA$_{0.1\%}$@U$_{60}$, 2NPAC$_{0.1\%}$@U$_{60}$, aNA$_{0.1\%}$@U$_{60}$, 3ABBA$_{0.1\%}$@U$_{60}$, respectively.

### Stability of TAPB$_{0.1\%}$@U$_{30}$ in solvents
To evaluate the stability of TAPB$_{0.1\%}$@U$_{30}$, it was dispersed in ethanol, 1,4-dioxane, tetrahydrofuran (THF), N,N-dimethylformamide (DMF), dimethyl sulfoxide (DMSO), ethyl acetate (EA), acetone, acetonitrile (ACN), dichloromethane (DCM), and even in aqueous medium at different pH values, respectively. The obtained suspensions were ultrasound for 30 min and then the RTP was recorded after turning off the UV lamp.

### Extraction of TAPB-N
TAPB$_{1.0\%}$@U$_{30}$ (1 g) was ultrasonically dissolved in 3 mL DMSO, 20 mL deionized water was added into the solution, the generated crystal powders were collected by filtration. The resulting solid powder was washed three times with deionized water, and then extracted with methanol (20 mL). The extracted solution was concentrated to yield TAPB-N.

### Preparation of PVA@TAPB and PVA@TAPB-N film
TAPB or TAPB-N (3 mg) was added to the PVA (100 mg) water solution (3 mL), it was stirred at room temperature until completely dissolved. Subsequently, the obtained aqueous solution was transferred to a watch glass and placed in an oven at 80 °C for 5 h.

## Data availability
All the other data used in this study are available in the article and its supplementary information files and from the corresponding author upon request.

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

## Acknowledgements

This work was financially supported by National Natural Science Foundation of China [grant numbers 21978208, Y.L.]; TianHe Qingsuo open research fund of TSYS in 2022 & NSCC-TJ [P-THQS-22-PY-No.0002, B.W.W.].

## Author contributions

Q.L.J.: conceptualized, investigation, formal analysis, validation, writing - original draft, writing - review & editing. X.L.Y.: supervision, project administration, writing - review & editing. B.W.W.: supervision, project administration. J.Y.L.: methodology, formal analysis. W.S.X.: data curation, formal analysis. Z.Y.S.: data curation, formal analysis. C.C.B.: data curation, project administration. Y.L.: data curation, project administration. L.G.C.: supervision, project administration.

## Competing interests

The authors declare no competing interests.
