## [Peer Review File · Nature Communications]

Construction of room temperature phosphorescent materials with ultralong lifetime by in-situ derivation strategyREVIEWER COMMENTS

Reviewer #1 (Remarks to the Author):

[Note from the Editor: Please also see attached PDF]

The authors doped the guest with amino or carboxyl group into urea, and the materials prepared by heating have excellent room temperature phosphorescence properties, and the afterglow time is up to 40 seconds. What is more precious is that doped materials have extremely strong corrosion resistance, and still have excellent RTP properties in common organic solvents or acid water. In general, this work has certain novelty. However, the writing logic of the paper is very chaotic, and it is difficult for this reviewer to quickly understand the content of the manuscript, for example, the abbreviations TAPB and U60 appearing in the text must be marked with their full names when they first appear. In addition, this reviewer did not understand which component in the material absorbed the energy of the excitation source and finally emitted phosphorescence. This reviewer thinks that the weakest part of the manuscript is the description of the mechanism of luminescent materials. The authors believe that the acid or amine produced in the pyrolysis of urea reacts with the carboxyl or amino group of the guest to generate the amide groups. However, the author has neglected a basic chemical knowledge. There is a great difference between the nitrogen atom and the carbon atom in the amide group, however, it is obvious that the authors believe that the guest connection on the carbon atom or the nitrogen atom has a similar effect. The theoretical calculation content in the manuscript is relatively simple, and there is no evidence of hydrogen bond formation of materials, et. al. To sum up, this reviewer thought it was a valuable work, but could not agree that the manuscript should be published at the current level.

Reviewer #2 (Remarks to the Author):

In this work, chen et al. developed a series of room temperature phosphorescence materials with ultra-long lifetime. The lifetime can reach up to 5.33 s with a 48 s afterglow, which is highly intriguing results. The phosphors in urea pyrolysis strategy supplied a new perspective for the design of ultra-long afterglow. However, the eventual pyrolysis product is a mixed composite, the efficient phosphorescent component is confusing, and related mechanism is unclear. Unfortunately, the results are not very convincing, and there are some errors in the manuscript and more supplements data should be provided. Based on these considerations, I expect this work should be reported in a more specialized journal in the future with more comprehensive study.

1. Since the isocyanic acid was generated during urea pyrolysis, and the cyan groups are the functional groups to generate cluster-triggered phosphorescence (Small, 2016, 12, 6586–6592), cluster-triggered RTP from isocyanic acid should be excluded.
2. In the experiment sections, P2BA and 3ABBA were heated with urea at 220 °C. However, boric acid groups are easy to dehydrate and crosslink at high temperature (Chem. Sci., 2017, 8, 8336–8344), the RTP from those triphenyl boroxine should be excluded.
3. Delayed time should be added into all the delayed spectra. Fluorescence emission still exists in

delayed emission in Figure 2C, does the delayed fluorescence emit in composite systems?

4. Figure 1 and Figure S1 should be mentioned in manuscript.

5. In Figure S2, 0.1 ms delayed time is short since the lifetime of TAPB0.1%@U30 up to 3.57 s. Because of the multiple components after urea pyrolysis, time-resolved spectra of TAPB0.1%@U30, delayed spectra (RT and 77 K) of pure U30 should be measured.

6. 5.33 s RTP lifetime is excited, but the fitted lifetime results exceed the real value since the decay curve is not attenuated completely by μs lamp. Kinetics decay measurement is more suitable.

7. In Figure S10, only one absorption band (350 nm) was observed, why did the author use 254, 312, and 365 nm excitation wavelength for the measurement of the spectra and decay curve lifetime?

8. From Figure S4-S6, "The above results indicated that the pyrolysis of urea underwent a urea-biuret-ammelide transformation pathway (Fig. 3)". How did the urea-biuret-ammelide transformation prove by SEM? Characteristic peaks should be marked in FT-IR and Raman spectra.

9. C=C bond doesn't show in Figure 3, why was the combined bond energy of C=C marked in Figure S6b?

10. It is an interesting phenomenon that the different luminescence was observed in different solvent and different pH values in Figure S7 and S8, the author should provide some explanation?

11. At line 168, "attributed to the fact that derivatives with abundant lone pair electrons were easier to transition to the excited state.", the related references should be added.

12. The decay curve lifetime of TAPB0.01@U30 in Figure S12b is different from Figure S14, but gained the same fitted lifetime 1.99 s. The authors should check the data.

13. The full name of the compound should be provided when they were mentioned for the first time. Such as TAPB at line 67, et al.

14. The whole manuscript should be double-checked by authors. "60 min. exhibited" at line 185, "TAPB0.1%@U30" at line 328, "The ground state (S_0)" at line 324.

15. It is hard to identify the data in the picture, higher resolution picture should be provided.

16. The format of references need to be unified, such as ref 11-14, 16,20, 31, 33, 39, and the DOI of ref 28 needs to be rechecked.

Reviewer #3 (Remarks to the Author):

In this Manuscript, the Authors report on a innovative method to build up organic materials showing ultralong room temperature phosphorescence (RTP). The method is based on incorporation of suitable guest molecules in urea, such as tris(aminophenyl)benzene (TAPB), followed by pyrolysis of the host-guest system allowing to generate guest derivatives characterized by phosphorescent emission. Incorporation of such emitters in the rigid urea matrix allows to obtain impressive ultralong RTP lifetimes.

The subject is of interest for the large community of scientists involved with the development of new organic materials with RTP features. The reported results are excellent and highly documented. However, at the present stage, the proposed mechanism underlying the observed phenomena is not sufficiently documented. The guest derivatives are only hypothesized on the basis of HRMS and their aggregation features (π - π stacking interactions, hydrogen bonding) are only guessed but not demonstrated. In my opinion, a more rigorous investigation on such 'intermediates' is needed to fully explain the obtained results.

Minor observations:

- TAPB acronym should be clarified in its first occurrence
- English should be improved

Response to the Reviewers' comments

Reviewer #1 (Remarks to the Author):

The authors doped the guest with amino or carboxyl group into urea, and the materials prepared by heating have excellent room temperature phosphorescence properties, and the afterglow time is up to 40 seconds. What is more precious is that doped materials have extremely strong corrosion resistance, and still have excellent RTP properties in common organic solvents or acid water. In general, this work has certain novelty. However, the writing logic of the paper is very chaotic, and it is difficult for this reviewer to quickly understand the content of the manuscript, for example, the abbreviations TAPB and U60 appearing in the text must be marked with their full names when they first appear. In addition, this reviewer did not understand which component in the material absorbed the energy of the excitation source and finally emitted phosphorescence. This reviewer thinks that the weakest part of the manuscript is the description of the mechanism of luminescent materials. The authors believe that the acid or amine produced in the pyrolysis of urea reacts with the carboxyl or amino group of the guest to generate the amide groups. However, the author has neglected a basic chemical knowledge. There is a great difference between the nitrogen atom and the carbon atom in the amide group, however, it is obvious that the authors believe that the guest connection on the carbon atom or the nitrogen atom has a similar effect. The theoretical calculation content in the manuscript is relatively simple, and there is no evidence of hydrogen bond formation of materials, et. al. To sum up, this reviewer thought it was a valuable work, but could not agree that the manuscript should be published at the current level.

Reply: Thanks very much for your consideration. The valuable comments are quite helpful for us to revise our manuscript. We have carefully considered all your comments and made revisions accordingly. Please refer to the detailed responses and modifications made in the manuscript. We hope that the corrections will meet with approval. Thanks again for your careful work on our manuscript.

- 1.** The writing logic of the paper is very chaotic, and it is difficult for this reviewer to quickly understand the content of the manuscript, for example, the abbreviations TAPB and U60 appearing in the text must be marked with their full names when they first appear.

Reply: Thank you for pointing out this problem. We apologize for our poor language skills. Therefore, we have checked the manuscript throughout strictly and revised our manuscript carefully, in which the subheadings were added to ensure the logical and clarity of expression.

Meanwhile, English language has been further polished, and other clerical errors including the problem of abbreviations you mentioned have been corrected (Page 4, Line 57; Page 5, Line 81; Page 10, Line 136; Page 13, Line 199). All revisions have been marked. Thanks again for your careful work on our manuscript.

2. This reviewer did not understand which component in the material absorbed the energy of the excitation source and finally emitted phosphorescence.

Reply: Thanks for your valuable comment. Sorry for the inconvenience to your review caused by our poor language expression. According to your suggestion, more characterization and theoretical calculations have been added in the revised manuscript, which further prove that a series of derivatives of guest molecules are the source of phosphorescent excitation and emission. TAPB was chosen as the template molecule for investigation. Firstly, the possibility of matrix luminescence was ruled out by comparing the photophysical properties of TAPB_{0.1%}@U₃₀ and U₃₀ (Page 6, Lines 86-93). Secondly, systematic characterizations indicated that a series of derivatives (TAPB-N) were generated in situ during the co-pyrolysis of urea and TAPB (Page 7, Lines 106-115). Finally, both theoretical calculations and control experiments showed that derivatives (TAPB-N) had similar RTP emission wavelengths to that of TAPB, and they worked together to achieve high performance RTP (Page 10, Lines 150-157). Thanks again for this helpful comment.

3. This reviewer thinks that the weakest part of the manuscript is the description of the mechanism of luminescent materials.

Reply: Thank you for your pointing out this problem. We apologize for inaccurate description of the emission mechanism. Conventional luminescent molecules, such as TAPB, are excited to yield triplet excitons *via* ISC process, which emit intense phosphorescence in solution at low temperature (Nat. Rev. Mater., 2020, 5, 869–885; Acc. Chem. Res., 2019, 52, 3, 738–748). In our work, taking TAPB as a template molecule, “in situ derivation” strategy was developed to enhance RTP emission by facilitating ISC processes and stabilizing triplet excitons. In the revised manuscript, more theoretical calculation and experimental results have been added and the luminescence mechanism of the material was thoroughly discussed.

In brief, the mechanism of luminescent materials could be described as follows: firstly, the derivatives were generated by the reaction between guest molecule and isocyanic acid or amine during the co-pyrolysis process (Page 7, Lines 106-115). Theoretical calculations showed that these derivatives exhibited different excited state properties from guest molecule, resulting in a

large SOC constant and a small ΔE_{ST} . This greatly increased the efficiency of phosphorescence emission by facilitating ISC (Page 10, Lines 134-160). Secondly, the rigid matrix formed by urea pyrolysis provided a good rigid environment for guest molecules, effectively reducing the non-radiative transition of triplet excitons and allowing them to release energy back to the ground state mainly in the form of phosphorescence (Page 11, Lines 161-176). The combination of derivatives and rigid matrix has an important contribution to high-performance RTP. Thanks again for this helpful comment.

4. The authors believe that the acid or amine produced in the pyrolysis of urea reacts with the carboxyl or amino group of the guest to generate the amide groups. However, the author has neglected a basic chemical knowledge. There is a great difference between the nitrogen atom and the carbon atom in the amide group, however, it is obvious that the authors believe that the guest connection on the carbon atom or the nitrogen atom has a similar effect.

Reply: Thanks for your comments. As the reviewer mentioned, there is a great difference between the nitrogen atom and the carbon atom in amide group. However, they exhibit similar effects on guest molecules in the "in situ derivation" strategy. For the whole guest molecule, the carbonyl group or amino group can change the density of the electron cloud in the surrounding region due to its electron withdrawing or electron donating properties. This can affect the electron excitation properties of guest molecules and effectively promote the separation of excited electrons and holes. To provide more evidence, detailed theoretical calculations were performed and supplemented in the revised manuscript. Specifically, TCPB and its derivatives showed similar calculation results to TAPB and its derivatives, further confirming the similar roles of carbonyl and amino groups in the "in situ derivation" strategy (Page 10, Lines 134-145; Page 15, Lines 225-230; Supplementary Analysis 3). Thanks again for your careful work.

5. The theoretical calculation content in the manuscript is relatively simple

Reply: Thanks for pointing out this problem. In the revised manuscript, we conducted more extensive and in-depth theoretical calculations based on the obtained results. Specifically, hole-electron analysis was performed to investigate the electronic properties of the excited state, because it considered all orbital transitions and could fully demonstrate the characteristics of electronic excitation (Page 10, Lines 134-145; Supplementary Figs. 13 and 23) (Carbon, 2020, 165, 461-467; Nat. Mater., 2020, 19, 1332-1338; J. Comput. Chem., 2012, 33, 580-592). In addition to hole and electron isosurface plots, we also introduced quantitative indicators for

hole-electron separation degree: the D index and t index, to ensure the results obtained are more persuasive (Supplementary Table 2). Moreover, the energy levels of various excited states of guest molecules and the spin-orbit coupling (SOC) constants between energy levels were calculated by ORCA, the results were consistent with those obtained from hole-electron analysis (Page 10, Lines 145-147; Supplementary Tables 3 and 6). The theoretical calculations also agree with our experimental results, providing stronger evidence for the "in situ derivation" strategy. Thanks again for this helpful comment.

6. There is no evidence of hydrogen bond formation of materials

Reply: Thanks for this valuable comment. In the revised manuscript, some characterization such as differential scanning calorimetry (DSC), Raman spectra and ^1H NMR spectra were supplemented to further confirm the hydrogen bonds in RTP materials. These characterization methods of hydrogen bonds have also been widely reported in previous studies, which provides strong evidence for our work (J. Am. Chem. Soc. 1998, 120, 4094-4104; Angew. Chem. Int. Ed., 2021, 60, 17094–17101; J. Raman Spectrosc., 2010, 41, 1708–1713; J. Raman Spectrosc., 2013, 44, 219–226). Detailed analysis has been supplemented in the revised manuscript (Page 11, Lines 162-173). This section significantly improved our work and thanks again for your helpful comment.

Reviewer #2 (Remarks to the Author):

In this work, Chen et al. developed a series of room temperature phosphorescence materials with ultra-long lifetime. The lifetime can reach up to 5.33 s with a 48 s afterglow, which is highly intriguing results. The phosphors in urea pyrolysis strategy supplied a new perspective for the design of ultra-long afterglow. However, the eventual pyrolysis product is a mixed composite, the efficient phosphorescent component is confusing, and related mechanism is unclear. Unfortunately, the results are not very convincing, and there are some errors in the manuscript and more supplements data should be provided. Based on these considerations, I expect this work should be reported in a more specialized journal in the future with more comprehensive study.

Reply: Thanks for your careful work on our manuscript. We appreciate your valuable suggestions, which have helped us improve the quality of the paper. In the section of material characterization, multiple characterizations related to rigid matrix and guest derivatives have been added (Page 7, Lines 108-115; Page 11, Lines 162-173). In the section of mechanism exploration, more detailed theoretical calculations were performed to explore the mechanism of RTP emission (Page 10, Lines 133-153). According to your suggestion, the original manuscript has been revised accordingly. Please refer to the detailed responses to each comment along with the revisions made in the manuscript. We hope that the corrections will meet with approval. Thanks again for your helpful comments.

1. Since the isocyanic acid was generated during urea pyrolysis, and the cyan groups are the functional groups to generate cluster-triggered phosphorescence (Small, 2016, 12, 6586–6592), cluster-triggered RTP from isocyanic acid should be excluded.

Reply: Thank you for pointing out this problem. In the revised manuscript, we added the photophysical data of U₃₀ derived from urea pyrolysis (Page 6, Lines 86-93). By comparing the photophysical data of U₃₀ and TAPB_{0.1%}@U₃₀, it was confirmed that the room temperature phosphorescence in our work was not cluster-triggered RTP from isocyanic acid. Additionally, isocyanic acid is a quite reactive compound that readily reacts with amines (Chem. Eng. J., 2009, 150, 544-550; Thermochem Acta, 2004, 424, 131-142). Thus, the isocyanic acid produced by pyrolysis will react quickly with the amines in the system and the possibility of isocyanic acid cluster-induced RTP can be excluded. A detailed discussion has been added in the revised manuscript. Thanks again for your careful work on our manuscript.

2. In the experiment sections, P2BA and 3ABBA were heated with urea at 220 °C. However, boric acid groups are easy to dehydrate and crosslink at high temperature (Chem. Sci., 2017, 8, 8336–8344), the RTP from those triphenyl boroxine should be excluded.

Reply: Thanks for your professional suggestion. Delayed PL spectra of P2BA_{0.1%}@U₆₀ and P2BA solution were compared, as shown in the figure below. The typical emission peak at 418 nm for the dilute solution of P2BA was consistent with the phosphorescence emission peak of P2BA_{0.1%}@U₆₀. Therefore, it can be confirmed that the phosphorescence emission of P2BA_{0.1%}@U₆₀ originates from the P2BA rather than those triphenyl boroxine (Nat. Photonics, 2019, 13, 406–411; Nat. Mater., 2021, 20, 1539–1544). Detailed discussions have been added in the revised manuscript (Page 13, Lines 207-210). Thanks again for your consideration on our manuscript.

3. Delayed time should be added into all the delayed spectra. Fluorescence emission still exists in delayed emission in Figure 2C, does the delayed fluorescence emit in composite systems?

Reply: Thanks for your suggestion. We apologize for the incomplete descriptions in the original manuscript. According to your opinion, the corresponding delay time of all the delayed emission spectra has been marked accordingly (Figs. 2 and 4; Supplementary Figs. 3, 4, 5, 14, 19, 25 and 27). Furthermore, a variable temperature phosphorescent spectroscopic test was conducted to evaluate the presence of delayed fluorescence in the obtained composites in this paper. The results showed that TCPB_{0.1%}@U₆₀ and 23NDCA_{0.1%}@U₆₀ displayed delayed fluorescence, while TAPB_{0.1%}@U₃₀ and P2BA_{0.1%}@U₆₀ did not, shown in the following figure.

Detailed discussions have been added in the revised manuscript and thanks again for your careful work on our manuscript. (Page 5, Line 81; Page 13, Line 210; Page 15, Line 228).

4. Figure 1 and Figure S1 should be mentioned in manuscript.

Reply: Thanks for your careful work. We apologize for the incomplete descriptions in original manuscript. Figure 1 and Figure S1 have been labeled in their correct positions, respectively (Page 4, Lines 64 and 66). Thanks again for your careful work on our manuscript.

5. In Figure S2, 0.1 ms delayed time is short since the lifetime of TAPB_{0.1%}@U₃₀ up to 3.57 s. Because of the multiple components after urea pyrolysis, time-resolved spectra of TAPB_{0.1%}@U₃₀, delayed spectra (RT and 77 K) of pure U₃₀ should be measured.

Reply: Thanks for your valuable suggestion.

1) As shown in the following figure, the phosphorescent spectra of the obtained material were retested by extending the delay time to 1 ms, and the emission peak (519 nm) was not significantly different from the original spectrum.

2) Time-resolved spectra of TAPB_{0.1%}@U₃₀ and delayed spectra (RT and 77 K) of U₃₀ have been measured and supplemented in the revised manuscript (Supplementary Fig. 3). By comparing the photophysical data of U₃₀ with TAPB_{0.1%}@U₃₀, the luminescence center of RTP material is determined to be guest molecule and its derivatives. Further detailed discussions have also been added in the revised manuscript (Page 6, Lines 86-93). Thanks again for this helpful comment.

6. 5.33 s RTP lifetime is excited, but the fitted lifetime results exceed the real value since the decay curve is not attenuated completely by μ s lamp. Kinetics decay measurement is more suitable.

Reply: Thanks for this suggestion. The phosphorescence lifetime of P2BA_{0.1%}@U₆₀ was determined again by Kinetics decay measurement and added it into the revised manuscript (Page 13, Line 205; Fig. 4c). As shown in the following figure, P2BA_{0.1%}@U₆₀ still exhibited an ultra-long lifetime of 5.02 s, further confirming that the material has good RTP performance (Reply Table 1). Thanks again for this helpful comment.

Reply Table 1 Lifetime of P2BA_{0.1%}@U₆₀

$\lambda_{em}(nm)$	$\tau_1(s)$	$\tau_2(s)$	$\tau_{av}(s)$
417	3.44(37.23%)	5.95(62.76%)	5.02

7. In Figure S10, only one absorption band (350 nm) was observed, why did the author use 254, 312, and 365 nm excitation wavelength for the measurement of the spectra and decay curve lifetime?

Reply: Thanks for pointing out this problem. In the original manuscript, we attempted to investigate the effects of excitation wavelengths on phosphorescence emission, ignoring the fact that the excitation spectrum of the material only showed one characteristic peak. Considering 365 nm is a common excitation wavelength of commercially available UV lamps, which is close to the unique absorption wavelength of 350 nm of the material, the data of 365 nm excitation wavelength is retained in the revised manuscript (Fig. 2c). Thanks again for this helpful comment.

8. From Figure S4-S6, “The above results indicated that the pyrolysis of urea underwent a urea-biuret-ammelide transformation pathway (Fig. 3)”. How did the urea-biuret-ammelide transformation prove by SEM? Characteristic peaks should be marked in FT-IR and Roman spectra.

Reply: Thanks for your helpful comment. We apologize for the inappropriate descriptions in the original manuscript. According to your suggestion, the characteristic peaks of FT-IR and Roman spectra were marked (Supplementary Fig. 7 and Supplementary Fig. 21). Considering that SEM can only observe the morphological changes of materials, the discussion on SEM characterization results is revised properly (Page 8, Line 121). Thanks again for your careful work on our manuscript.

9. C=C bond doesn't show in Figure 3, why was the combined bond energy of C=C marked in Figure S6b?

Reply: Thanks for pointing out this problem. We feel very sorry for this mistake. The binding energy of 284.8 eV was attributed to the C_{1s} peak (C-C) of the surface contamination layer rather than C=C structure (ISO 19318:2004; Sci Rep, 2021, 11, 11195), which has been corrected in the revised manuscript (Supplementary Fig. 8). Thanks again for your careful work on our manuscript.

10. It is an interesting phenomenon that the different luminescence was observed in different solvent and different pH values in Figure S7 and S8, the author should provide some explanation?

Reply: Thanks for your suggestion. Discussion on the effects of solvent and pH on their luminescent properties has been added in the revised manuscript (Page 12, Lines 176-183). Thanks again for this helpful comment.

11. At line 168, “attributed to the fact that derivatives with abundant lone pair electrons were easier to transition to the excited state.”, the related references should be added.

Reply: Thanks for your suggestion. The original sentence has been modified to “Compared with TAPB, the red shift of the main emission peaks in the TAPB_{0.1%}@U₃₀ delayed PL and excitation spectra might be attributed to the derivatization of TAPB and the domain limiting effect of hydrogen bond network on guest derivatives in matrix” (Page 7, Line 93). The related references have been added in the revised manuscript (Reference 33-35). Thanks again for this helpful comment.

12. The decay curve lifetime of TAPB_{0.01%}@U₃₀ in Figure S12b is different from Figure S14, but gained the same fitted lifetime 1.99 s. The authors should check the data.

Reply: Thanks for this important comment. As mentioned by the reviewer, the decay curve lifetime of TAPB_{0.01%}@U₃₀ was presented both in Supplementary Fig. 5c (original Figure S12b) and Supplementary Fig. 12 (original Figure S14). The same data was used for the decay curve lifetime of TAPB_{0.01%}@U₃₀ in these two figures, resulting in the same fitting result. Due to our carelessness, the decay curve of TAPB_{0.01%}@U₃₀ in Supplementary Fig. 12 was shifted upward a little when it was processed in Origin software, which has been corrected in the revised manuscript. We apologize for this mistake and really appreciate your careful work.

13. The full name of the compound should be provided when they were mentioned for the first time. Such as TAPB at line 67, et al.

Reply: Thank you for pointing out this problem. We feel sorry for our carelessness. The full names of the abbreviations have been marked when they were mentioned for the first time in the revised manuscript (Page 4, Line 57; Page 5, Line 81; Page 10, Line 136; Page 13, Line 199). All revisions have been marked. Thanks again for your careful work on our manuscript.

14. The whole manuscript should be double-checked by authors. “60 min. exhibited” at line 185, “TAPB0.1% @U30” at line 328, “The ground state (S0)” at line 324.

Reply: Thanks for your suggestion. We feel sorry for these mistakes. We have carefully checked the manuscript and tried our best to revise the writing norms. All revisions have been marked (Page 4, Line 57; Page 8, Line 125; Page 10, Line 146; Page 19, Line 307; Page 19, Line 309). Thanks again for your careful work on our manuscript.

15. It is hard to identify the data in the picture, higher resolution picture should be provided.

Reply: Thank you for this suggestion. Higher resolution pictures were provided in revised manuscript. Furthermore, considering the compression of image quality by text editing software, we have already uploaded the high-resolution images in a compressed package when uploading the revised manuscript. Thanks again for your careful work on our manuscript.

16. The format of references needs to be unified, such as ref 11-14, 16,20, 31, 33, 39, and the DOI of ref 28 needs to be rechecked.

Reply: Thanks very much for pointing out this problem. The inappropriate format of references has been corrected in the revised manuscript according to the reference format requirements of Nature Communications. Thanks again for your careful work on our manuscript.

Reviewer #3 (Remarks to the Author):

In this Manuscript, the Authors report on an innovative method to build up organic materials showing ultralong room temperature phosphorescence (RTP). The method is based on incorporation of suitable guest molecules in urea, such as tris(aminophenyl)benzene (TAPB), followed by pyrolysis of the host-guest system allowing to generate guest derivatives characterized by phosphorescent emission. Incorporation of such emitters in the rigid urea matrix allows to obtain impressive ultralong RTP lifetimes.

The subject is of interest for the large community of scientists involved with the development of new organic materials with RTP features. The reported results are excellent and highly documented. However, at the present stage, the proposed mechanism underlying the observed phenomena is not sufficiently documented. The guest derivatives are only hypothesized on the basis of HRMS and their aggregation features (pi-pi stacking interactions, hydrogen bonding) are only guessed but not demonstrated. In my opinion, a more rigorous investigation on such 'intermediates' is needed to fully explain the obtained results.

Minor observations:

- TAPB acronym should be clarified in its first occurrence
- English should be improved

Reply: Thanks for your appreciation and recognition of our manuscript. We also greatly appreciate you for the very careful review, and our manuscript has been improved according to your suggestion. In the revised manuscript, we have conducted a more detailed exploration of guest derivatives and improved the English of the manuscript. We hope that these changes can gain your recognition and contribute to the acceptance of the article.

1. In my opinion, a more rigorous investigation on such 'intermediates' is needed to fully explain the obtained results.

Reply: Thanks very much for your consideration. In the revised manuscript, we have supplemented new characterizations to ensure more rational and accurate identification of derivatives. Specifically, the derivatives were further characterized by Raman spectrum and ¹H NMR (Page 7, Lines 108-116). In order to better study the electronic excitation properties of the derivatives, more detailed theoretical calculations have been carried out, and the results of calculations were further verified by control experiments (Page 9, Lines 132-160). These results

indicate that the derivatives can effectively promote the ISC process and increase the phosphorescent quantum yield. This further demonstrated the effectiveness of “in situ derivation” strategy in constructing high-performance RTP materials. A more detailed explanation has been added in the revised manuscript. All revisions have been marked. Thanks again for your careful work on our manuscript.

2. TAPB acronym should be clarified in its first occurrence

Reply: We appreciate for your careful check. In the revised manuscript, the full names of the abbreviations have been marked in its first occurrence (Page 4, Line 57; Page 5, Line 81; Page 10, Line 136; Page 13, Line 199). We feel sorry for our carelessness and thanks again for pointing out this problem.

3. English should be improved

Reply: Thanks for your valuable suggestion. We have carefully checked the manuscript and tried our best to polished English. All revisions have been marked. Thanks again for your careful work on our manuscript.

REVIEWERS' COMMENTS

Reviewer #1 (Remarks to the Author):

The revised manuscript can be published.

Reviewer #2 (Remarks to the Author):

The resubmitted manuscript has been greatly improved. The authors have addressed the issues and made the requested modification. In my opinion, the paper is now suitable for publication in Nature Communicaitons.

Reviewer #3 (Remarks to the Author):

The Manuscript has considerably improved compared to the original version. All the criticisms raised by my previous review have been addressed. I deem that now the Manuscript is suitable for publication on Nature Communications.

Dear Editor,

Thank you for your kind consideration and reviewers' valuable comments on our manuscript (reference number: NCOMMS-23-04660B). Now we have revised our paper carefully to comply with the policies and formatting requirements of Nature Communications. The revised portions are marked by blue in the revised manuscript. The point-to-point response to reviewers' comments are listed as following.

Reviewer #1 (Remarks to the Author):

The revised manuscript can be published.

Reply: Many thanks for your kind recommendation!

Reviewer #2 (Remarks to the Author):

The resubmitted manuscript has been greatly improved. The authors have addressed the issues and made the requested modification. In my opinion, the paper is now suitable for publication in Nature Communications.

Reply: Thanks for your kind recommendation of this work!

Reviewer #3 (Remarks to the Author):

The Manuscript has considerably improved compared to the original version. All the criticisms raised by my previous review have been addressed. I deem that now the Manuscript is suitable for publication on Nature Communications.

Reply: We appreciate your consideration!

In the end, we appreciate the editor and reviewers' warm work earnestly.

Yours sincerely,

Ligong Chen